# MMFNet: A Mamba-Based Multimodal Fusion Network for Remote Sensing Image Semantic Segmentation

**DOI:** 10.3390/s25196225

**Published:** 2025-10-08

**Authors:** Jingting Qiu, Wei Chang, Wei Ren, Shanshan Hou, Ronghao Yang

**Affiliations:** 1College of Earth and Planetary Sciences, Chengdu University of Technology, Chengdu 610059, China; 2023020174@stu.cdut.edu.cn; 2Qi-Liang Spatiotemporal Information Innovation Studio, The 9th Geological Brigade of Sichuan Province, Deyang 618029, China; 18628115380@163.com (W.R.); sshou341@163.com (S.H.); 3Geographic Information and Geological Environment Key Laboratory of Deyang City, Deyang 618029, China

**Keywords:** multimodal semantic segmentation, remote sensing, feature fusion

## Abstract

Accurate semantic segmentation of high-resolution remote sensing imagery is challenged by substantial intra-class variability, inter-class similarity, and the limitations of single-modality data. This paper proposes MMFNet, a novel multimodal fusion network that leverages the Mamba architecture to efficiently capture long-range dependencies for semantic segmentation tasks. MMFNet adopts a dual-encoder design, combining ResNet-18 for local detail extraction and VMamba for global contextual modelling, striking a balance between segmentation accuracy and computational efficiency. A Multimodal Feature Fusion Block (MFFB) is introduced to effectively integrate complementary information from optical imagery and digital surface models (DSMs), thereby enhancing multimodal feature interaction and improving segmentation accuracy. Furthermore, a frequency-aware upsampling module (FreqFusion) is incorporated in the decoder to enhance boundary delineation and recover fine spatial details. Extensive experiments on the ISPRS Vaihingen and Potsdam benchmarks demonstrate that MMFNet achieves mean IoU scores of 83.50% and 86.06%, outperforming eight state-of-the-art methods while maintaining relatively low computational complexity. These results highlight MMFNet’s potential for efficient and accurate multimodal semantic segmentation in remote sensing applications.

## 1. Introduction

Remote sensing images (RSIs) serve as a vital data source for Earth observation. Semantic segmentation of RSIs, also referred to as land use and land cover (LULC) classification [1], aims to assign a semantic label to each pixel, enabling pixel-level categorization. LULC classification plays a crucial role in various applications, including urban planning [2], agricultural management [3] environmental monitoring [4], and the development of geographic information system (GIS) [5].

High-resolution remote sensing images (HRRSIs) provide abundant texture, geometric, and spectral information [6], yet their diversity and complexity pose significant challenges for semantic segmentation. Large intra-class variance, small inter-class variance, and shadow interference reduce class separability and hinder accurate feature extraction [7]—for instance, the difficulty in distinguishing between trees and low vegetation in the first row of Figure 1. Moreover, shadows in HRRSIs can obscure important features, as illustrated in the second row of Figure 1, where shadowing impedes the identification of a path between tree canopies. These challenges highlight the need for robust segmentation methods capable of capturing both fine-grained local details and long-range contextual information under complex imaging conditions, which forms the primary motivation of this study.

Compared with unimodal approaches, multimodal semantic segmentation (MSS) integrates diverse remote sensing sources—such as digital surface models (DSMs) and multispectral, panchromatic, and radar data—to exploit complementary information across modalities, thereby improving segmentation accuracy [8]. For instance, DSMFNet [9] introduces a DSMF module that extracts informative features from DSM data to enhance segmentation performance in shadowed regions or areas with similar colour and texture. CMGFNet [10] employs a Gated Fusion Module (GFM) to adaptively combine features from different modalities while suppressing redundancy. Similarly, CMFNet [11] uses a cross-attention mechanism to fuse multimodal features at multiple scales, achieving cross-modal integration with multiscale awareness.

Most existing multimodal segmentation networks, however, rely on convolutional neural networks (CNNs) [12] or vision transformer (ViT) [13] as backbone architectures. CNNs excel at extracting local features due to their local connectivity and weight sharing, but their limited receptive field constrains their ability to model global context [14]. ViTs [13], on the other hand, offer stronger long-range dependency modelling, but their quadratic computational complexity hinders their applicability to high-resolution imagery [15]. Thus, multimodal segmentation models based on CNNs or ViTs face a fundamental trade-off between capturing global context and maintaining computational efficiency.

Mamba [16], a variant of the State Space Model (SSM), was originally introduced for natural language processing. It offers the capability to model long-range dependencies while maintaining linear computational complexity. Its successful adaptation to RSI segmentation, as demonstrated by Vim [17] and VMamba [18], highlights its potential as an alternative to transformer-based architectures. Sigma [19] represents the first attempt to apply SSMs to multimodal segmentation, proposing an end-to-end network entirely built upon the Mamba architecture. Specifically, Sigma [19] adopts VMamba as the feature extraction backbone and designs two Mamba-based fusion modules, Cross Mamba Block (CroMB) and Concat Mamba Block (ConMB), to facilitate cross-modal interaction. Despite Mamba’s strength in capturing long-range dependencies, it performs suboptimally in segmenting small-scale objects. To address this limitation, MFMamba [20] combines CNN and VMamba backbones to separately extract multimodal features. It incorporates a Feature Fusion Block (FFB) to enhance both global and local modality-specific features, which are then aggregated via element-wise addition. While this approach effectively captures multi-scale global and local features, it remains susceptible to information loss and feature misalignment, particularly in complex remote sensing scenes with large inter-modal differences.

In summary, several challenges persist in MSS of RSI:

(1) Balancing accuracy and efficiency. MSS is a dense, pixel-level classification task. Current approaches often adopt dual-encoder frameworks based on CNNs and Transformers to capture local and global semantic features, respectively. However, Transformer-based methods incur substantial computational overhead, making it difficult to achieve both high accuracy and computational efficiency.

(2) Effective multimodal feature fusion. Most existing methods perform feature fusion through concatenation or element-wise addition. However, when significant differences exist between modalities, such strategies may lead to information loss and feature misalignment. Moreover, redundant information may be introduced, amplifying the impact of image noise and ultimately degrading segmentation accuracy [21].

To address these challenges, we propose MMFNet, a multimodal dual-encoder fusion network based on the Mamba architecture. In this framework, the primary encoder adopts ResNet-18 [22] to extract local features, while the auxiliary encoder replaces the transformer with VMamba to capture global semantic features. It strikes a balance between accurate long-range dependency modelling and computational efficiency. To facilitate cross-modal interaction between the two encoders and to exploit complementary information across modalities, we introduce a multimodal feature fusion block (MFFB) at each feature extraction stage. This module integrates global and local information using multi-kernel convolution and window-based cross-attention, ensuring the effective fusion of multimodal features.

In the decoder stage, to better integrate deep and shallow features both semantically and in the frequency domain, we employ FreqFusion [23], a frequency-aware guided upsampling method, as a replacement for conventional feature fusion techniques. This approach enhances the restoration of fine spatial details and contributes to improved segmentation accuracy. In summary, the main contributions of this study are as follows:

A novel MFFB is designed to extract complementary features across modalities. The local branch captures fine-grained details using multi-scale depthwise separable convolutions, while the global branch employs Efficient Additive Attention [24] to model global contextual information. These are further integrated via window-based cross-attention to enable effective interaction between local and global representations.

A frequency-aware guided upsampling method (FreqFusion) is introduced in the decoder stage to replace traditional upsampling strategies. This facilitates semantic fusion of deep and shallow features and enables finer reconstruction of spatial details.

Extensive experiments conducted on the ISPRS Vaihingen and Potsdam datasets demonstrate that MMFNet achieves superior segmentation performance compared to eight state-of-the-art methods, while maintaining low computational complexity.

## 2. Related Work

### 2.1. Single Modal Semantic Segmentation

In recent years, deep learning has emerged as the dominant approach for semantic segmentation. Fully Convolutional Network (FCN) [25] marked a major milestone as the first end-to-end architecture for pixel-level prediction, significantly improving segmentation accuracy. However, FCNs rely solely on deep semantic features and perform a single-step upsampling, which often leads to blurred object boundaries and limited segmentation precision. To address these limitations, U-Net [26] introduced an encoder–decoder architecture with skip connections. This design enables the progressive fusion of deep semantic features and shallow spatial details, effectively refining object boundaries and enhancing segmentation accuracy.

Many researchers have enhanced the global context modelling capabilities of CNNs by stacking multiple convolutional layers, employing dilated convolutions [27], or introducing attention mechanisms [28]. However, dilated convolutions often fail to capture fine-grained details, thereby weakening the network’s ability to segment small objects. Attention mechanisms, while improving contextual awareness, still rely on convolutional backbones and thus remain inherently limited to local information exchange. Since convolution operations extract features within fixed receptive fields, CNN-based methods are fundamentally ineffective in capturing global semantic context and long-range dependencies [29]. Due to their powerful ability to model long-range visual dependencies, Transformers have attracted increasing attention in computer vision tasks. ViT, divides input images into patches and processes them using a pure Transformer architecture for image classification, achieving promising results. Compared to CNNs, ViT significantly improves the modelling of global contextual information, which has inspired a series of subsequent works [30,31,32,33].

However, the quadratic computational complexity of Transformers with respect to input size poses significant challenges when applied to dense prediction tasks such as object detection and semantic segmentation. This has created a pressing need for a novel vision backbone that offers global receptive fields and dynamic weighting while retaining linear complexity.

The recently proposed Mamba model has emerged as a promising alternative to Transformers, owing to its ability to model long-range dependencies with linear computational cost. Recent works such as Vim and VMamba have successfully extended Mamba to visual tasks, demonstrating its effectiveness in image representation learning.

In the context of RSI segmentation, Zhu et al. replaced the multi-head self-attention in ViT with Mamba to capture global dependencies from image data, and integrated it with multilayer perceptron (MLP) to construct the Samba block [34]. Additionally, RS-Mamba [35] introduces a VSS module equipped with an Omnidirectional Selective Scan Module (OSSM), which enhances multi-directional global context modelling and enables comprehensive spatial feature extraction.

### 2.2. Multimodal Semantic Segmentation (MSS)

With advances in Earth observation technologies, the acquisition of multimodal remote sensing data has significantly improved. Increasing attention has been given to the integration of data such as optical imagery, multispectral images, and DSMs. DSMs capture surface height information and serve as key indicators for identifying objects with similar spectral features but different elevations, such as trees and low-lying vegetation in confined areas. The high spatial precision of DSMs enhances boundary delineation, thereby contributing to improved segmentation performance in RSI.

Multimodal fusion strategies are generally classified into three categories: early fusion, intermediate fusion, and late fusion.

Early fusion involves stacking multiple modalities along the channel dimension before feeding them into the network. For instance, ResUNet-a [36] concatenates RGB images (three channels) and DSM data (one channel) into a four-channel input. However, this straightforward channel-wise concatenation fails to fully exploit the complementary nature of multimodal information, often introducing noise or redundancy. As a result, cross-modal feature extraction remains suboptimal, and segmentation accuracy gains are limited.

Late fusion, on the other hand, extracts feature from each modality using separate branches and combines their outputs at the decision level—typically via weighted averaging, voting, or ensemble classification. For example, V-FuseNet [37] utilizes two independent branches to process RGB and DSM data separately and fuses them by element-wise addition at the output layer. While this approach preserves modality-specific features, the lack of interaction during feature extraction leads to a loss of cross-modal correlations, limiting the effectiveness of the final prediction.

To overcome the limitations of early and late fusion, most recent studies favour intermediate fusion strategies. This approach typically employs parallel dual-branch networks to extract features from each modality independently, followed by hierarchical fusion at intermediate layers. The fused features are then upsampled using a decoder for final prediction.

FuseNet [38] uses two CNNs to extract features from RGB and depth images, respectively, and integrates depth features into the RGB stream at various levels using a sparse fusion strategy. CMGFNet [11] introduces a GFM to adaptively combine multimodal features while reducing redundancy during fusion. CMFNet [12] leverages a cross-attention mechanism to integrate multimodal information at multiple scales. EDFT [39], built upon SegFormer [32], adopts a two-branch architecture and proposes a depth-aware self-attention (DSA) module to fuse multimodal features. Intermediate fusion not only enables effective intra-modality feature extraction but also progressively aligns features across modalities by integrating information at multiple scales, thereby enhancing the overall representational power of multimodal data.

Mamba combines the ability to model long-range dependencies with the advantage of linear computational complexity, making it a compelling alternative to conventional Transformer-based architectures. Recent studies have explored its potential in multimodal learning tasks. AlignMamba [40] introduces a novel fusion framework that employs local alignment via optimal transport and global alignment via maximum mean discrepancy (MMD) to address the cross-modal misalignment challenge inherent in Mamba’s sequential processing. This enhances the efficiency of multimodal feature fusion.

Sigma [19] is the first to apply Mamba to MSS, proposing a Mamba-based cross-attention mechanism to facilitate interaction across modalities. In the field of joint classification of HSI (Hyperspectral Image) and LiDAR data, M2FMNet [41] achieves comprehensive feature fusion of hyperspectral and DSM data by introducing the Spatial-Spectral Adaptive Module (S2AM) and Elevation-Enhanced Module (E2M), thereby improving the accuracy of land cover classification. However, its fusion strategy mainly relies on pixel-level integration with residual connections, which limits its ability to capture fine-grained semantic boundaries and complex cross-modal interactions. In the context of Joint classification of visible image and DSM data, MFMamba [21] represents a pioneering effort. It utilizes CNNs to extract local features and Mamba to capture global context. However, its fusion strategy—element-wise addition of local and global features—limits the extent of cross-modal interaction and weakens the model’s ability to represent complex spatial relationships. Compared with MFMamba, MambaTriNet [42] achieves more comprehensive multimodal feature interaction through its TriFusion and CLFG modules. However, its three-branch design significantly increases computational complexity and structural burden, which limits its applicability in resource-limited scenarios.

The aforementioned studies demonstrate that although recent Mamba-based multimodal fusion networks have made progress in global feature modelling and cross-modal interaction, they still face a major challenge: how to simultaneously ensure computational efficiency and sufficiently effective cross-modal fusion in multimodal remote sensing semantic segmentation. To address this, our proposed MMFNet introduces lightweight Multi-Modal Feature Fusion Blocks (MFFB) and a frequency fusion (FreqFusion) strategy, which enhance spectral–structural complementarity while maintaining lower complexity, thereby achieving a better balance between accuracy and efficiency.

## 3. Methodology

### 3.1. Framework of MMFNet

As illustrated in Figure 2, MMFNet adopts a dual-encoder architecture followed by a decoder, designed for effective multimodal feature fusion and accurate semantic segmentation. The primary encoder is based on ResNet-18 to capture fine-grained local details, while the auxiliary encoder utilizes VMamba to efficiently model global context, achieving long-range dependency learning with linear computational cost. At each stage of the encoder, a Multimodal Feature Fusion Block (MFFB) integrates complementary features from both modalities. In the decoder, a frequency-aware upsampling module (FreqFusion) combines deep and shallow features to enhance spatial detail recovery. The entire network is optimized using a cross-entropy loss function.

### 3.2. Dual Branch Encoder

As shown in Figure 2, the primary encoder follows the standard ResNet-18 architecture, consisting of four residual stages for progressive local feature extraction. The auxiliary encoder (as shown in Figure 3) employs the Visual State Space (VSS) module, whose core Selective Scanning 2D (SS2D) mechanism enables efficient modelling of long-range dependencies while preserving computational efficiency. Both encoders downsample inputs hierarchically and extract multi-scale features, which are then passed to the MFFB for stage-wise multimodal integration. This dual-encoder design, as illustrated in Figure 2, allows MMFNet to effectively combine local detail sensitivity with global semantic awareness, thereby addressing the limitations of single-modality and single-backbone approaches.

The auxiliary encoder also comprises four stages. It performs hierarchical downsampling through patch embedding and patch merging operations, while the integrated VSS modules progressively learn global semantic representations across scales.

This architectural design integrates the strengths of CNNs in local feature extraction with the global semantic modelling capabilities of Mamba, enabling an effective mechanism for collaborative representation of multimodal features.

### 3.3. Multimodal Feature Fusion Block

Fusion methods based on simple concatenation or alignment often fail to fully exploit the complementary characteristics of cross modal data [43] while also neglecting the long-range dependencies between multimodal inputs [43]. Moreover, inherent noise and redundant features may further degrade the quality of feature representation [9].

To effectively model both intra-modal and inter-modal dependencies, and to extract modality-specific as well as modality-shared features, we design the MFFB. As illustrated in Figure 4, MFFB consists of three main components: a local branch, a global branch, and a window cross attention (WCA) module. The local and global branches are responsible for capturing the detailed or global features that may be lacking in certain modalities, while the WCA module models cross-modal dependencies to obtain complementary features across modalities.

The MFFB takes as input the feature maps *F_Ri_* from the CNN-based primary encoder and *F_Vi_* from the VSS-based auxiliary encoder. Due to the limited receptive field of CNNs, the feature maps *F_Ri_* from the primary encoder lack sufficient global context. Conversely, although Mamba excels at modelling long-range dependencies, it is less effective in capturing fine-grained local details, resulting in relatively coarse representations in *F_Vi_* from the auxiliary encoder. To integrate the global and local features from *F_Ri_* and *F_Vi_*, MFFB processes *F_Ri_* through the global branch, where Efficient Additive Attention (for details, see [24]) is applied to extract global contextual features *F_Global_*.

Simultaneously, *F_Vi_* is passed through the local branch, where three convolutional layers with different kernel sizes are used to capture fine-grained local features *F_Local_*. The outputs of the global and local branches are then fused via the WCA module, as illustrated in Figure 4.

To avoid the high computational cost of global attention, both *F_Local_* and *F_Global_* are first partitioned into non-overlapping windows of size 7 × 7. These windowed features are then passed through linear layers to generate the corresponding queries (*Q*), keys (*K*), and values (*V*), which are subsequently used to compute the cross attention. This process can be expressed by Equations (1)–(3) as:(1)$$Q=W_{Q}F_{Global},K=W_{K}F_{Local},V=W_{V}F_{Local}$$(2)$$Attention(Q,K,V)=\mathrm{SoftMax}(\frac{QK^{T}}{\sqrt{d}}+B)V$$(3)$$F_{out}=MLP(Attention(Q,K,V))$$
here, $B\in R^{W^{2}\times W^{2}}$ denotes the relative positional bias, while $Q,K,V\in R^{W^{2}\times d}$ represent the query, key, and value matrices, respectively; *d* refers to the dimensionality of the query/key vectors, and *W* is the window size.

The WCA model performs cross attention computation within each local window. By modelling cross modal dependencies in a windowed manner, WCA effectively captures complementary features across modalities with reduced computational overhead.

### 3.4. Transfomer Decoder

HRRSI poses significant challenges for object recognition and localization due to its high spatial resolution, dense object distribution, and wide variations in object scale. Therefore, the combination of global context and fine spatial detail is critical for reliable semantic reasoning in HRRSI [44]. Although encoder–decoder architectures such as U-Net integrate shallow high-resolution features with deep semantic representations via skip connections, their decoders typically rely on fixed interpolation kernels—such as Bilinear Interpolation or Nearest Neighbour Interpolation—which are insufficient for capturing the rich contextual information required for accurate semantic reasoning. To address this limitation, FreqFusion [23] introduces an adaptive frequency-domain upsampling strategy. Specifically, it applies adaptive low-pass filtering to upsampling deep (low-resolution) features by suppressing high-frequency noise and maintaining semantic consistency. A displacement generator is further used to perform spatial alignment. Meanwhile, adaptive high-pass filtering is applied to enhance boundary details in shallow (high-resolution) features, compensating for the loss of high-frequency information during downsampling. This enables complementary fusion of deep and shallow features in the frequency domain.

The decoder in this study adopts a hybrid architecture combining global local transformer block (GLTB) and FreqFusion. GLTB employs a multi-scale attention mechanism to simultaneously capture long-range dependencies and local spatial details. FreqFusion performs frequency-aware upsampling on deep (low-resolution) features and utilizes an adaptive frequency filtering mechanism in the spatial domain to achieve complementary fusion of deep and shallow features in terms of both semantic consistency and fine-grained detail. Further details of GLTB and FreqFusion can be found in [23,33].

### 3.5. Loss Function

We employ the cross-entropy loss to supervise the training of the network, which is defined as follows:(4)$$L_{CE}=-\sum_{i=1}^{n}t_{i}\log(p_{i})$$
where *t_i_* denotes the GT, and *p_i_* is the softmax probability of class *i*.

## 4. Experiment and Results

### 4.1. Dataset

(1) Vaihingen: Vaihingen dataset consists of 16 high-resolution true orthophoto images, each with an average size of approximately 2500 × 2000 pixels. Each image contains three spectral channels—near-infrared (NIR), red, and green (NIRRG)—as well as a DSM with a ground sampling distance (GSD) of 9 cm. The dataset includes five foreground classes: building (Bui.), tree (Tre.), low vegetation (Low.), car, and impervious surface (Imp.), as well as one background class: clutter. In our experiments, we utilized TOP image tiles and complete images. The 16 images are divided into a training set of 12 images and a test set of 4 images. The training set includes image IDs: 1, 3, 23, 26, 7, 11, 13, 28, 17, 32, 34, and 37; the test set comprises images 5, 21, 15, and 30.

(2) Potsdam: Potsdam dataset consists of 24 high-resolution aerial images captured over the city of Potsdam, Germany, each with a resolution of 6000 × 6000 pixels and a GSD of 5 cm. It provides four multispectral channels, including infrared, red, green, and blue (IRRGB), along with a DSM at the same 5 cm GSD. The dataset shares the same semantic classes as the Vaihingen dataset. In our experiments, we use the RGB composite images together with the corresponding DSM data. The dataset is divided into 18 images for training and 16 images for validation or testing. The training set comprises the following image IDs: 6_10, 7_10, 2_12, 3_11, 2_10, 7_8, 5_10, 3_12, 5_12, 7_11, 7_9, 6_9, 7_7, 4_12, 6_8, 6_12, 6_7, and 4_11. The test/validation set includes: 2_11, 3_10, 6_10, 7_10, 2_12, and 3_11. Figure 5 presents some data samples from the Vaihingen and Potsdam datasets.

### 4.2. Evaluation Metrics

To quantitatively assess the performance of segmentation, we adopt four commonly used metrics: Intersection over Union (IoU), mean IoU (mIoU), overall accuracy (OA), and mean F1-score (mF1). These metrics are computed based on four fundamental quantities: true positives (TP), true negatives (TN), false positives (FP), and false negatives (FN) represent, respectively, the number of pixels correctly classified as belonging to a class, correctly classified as not belonging to a class, incorrectly classified as belonging to a class, and incorrectly classified as not belonging to a class.

For each class, IoU is defined as the ratio between the intersection and the union of the predicted and GT regions, and is calculated as follows:(5)$$\mathrm{IoU}=\frac{\mathrm{TP}}{\mathrm{TP}+\mathrm{FP}+\mathrm{FN}}$$

The mIoU and mF1 for each class are calculated as follows (Equations (6) and (7)), and the F1-score is computed in the conventional manner as described in [21].(6)$$\mathrm{mIoU}=\frac{1}{c}\sum_{i=1}^{C}{\mathrm{IoU}}_{i}$$(7)$$\mathrm{mF}1=\frac{1}{C}\sum_{i=1}^{C}{F1}_{i}$$
where *i* denotes the class index and *C* is the total number of classes, mIoU refers to the mean Intersection over Union computed across all classes. Specifically, the IoU of each class is calculated using Equation (5), and the arithmetic mean of these values is reported as mIoU according to Equation (6). Similarly, mF1 denotes the macro-averaged F1-score across all classes, where the F1-score for each class is first calculated, and then averaged equally across all categories according to Equation (7).

### 4.3. Experiment Setup

All experiments were implemented using PyTorch2.2.2 and conducted on a single RTX 4080 GPU. During training, images were randomly cropped into 256 × 256 patches, and data augmentation techniques such as random horizontal flipping, random vertical flipping, and random rotation were applied. The number of training epochs was set to 50. The model was optimized using Stochastic Gradient Descent (SGD) with a learning rate of 0.01, a momentum of 0.9, a weight decay of 0.0005, and a batch size of 16.

### 4.4. Experimental Results and Analysis

#### 4.4.1. Comparison Results on the Vaihingen Dataset

As shown in Table 1, the proposed MMFNet achieves the highest scores on the Vaihingen dataset in terms of OA, mF1, and mIoU. Compared with the baseline MFMamba, MMFNet shows consistent improvements of 1.59% in mF1 and 2.04% in mIoU, demonstrating its ability to effectively extract and fuse complementary features from DSM and HRRSI data. In comparison with existing SOTA methods, MMFNet also delivers superior segmentation performance in key categories such as building, tree, low vegetation, and impervious surface. Specifically, it achieves an increase of 2.41% in IoU for low vegetation and 1.21% in IoU for car over the baseline, further highlighting its advantage in modelling fine-grained multimodal features.

Figure 6 presents visual comparisons of the segmentation results obtained by the nine methods. In the first and second rows, several single-modality models misclassify regions within buildings as impervious surfaces. In contrast, our method and CMFNet yield the most accurate building predictions. In the second row, the building contours predicted by our model are more consistent with the GT, exhibiting smoother and more complete edges. This demonstrates the ability of MMFNet to more precisely segment large-scale structures. Additionally, within the purple box in the second row, MMFNet successfully identifies trees located within areas of low vegetation, whereas other methods (Figure 6d–k) either miss or partially detect them. This result highlights MMFNet’s superior effectiveness in addressing the challenge of inter-class similarity, and the example in the top-left purple box of Figure 7a further validates this. In the third row of Figure 6, the purple box marks a region where shadows cast by trees on both sides of the road alter the appearance of the surface in RGB images, making the road’s colour and texture significantly different from the surrounding areas. As a result, most comparison methods misclassify the region. However, MMFNet accurately distinguishes the road from adjacent trees, demonstrating its robustness in mitigating the adverse effects of shadow occlusion. Similarly, in the region shown in the right purple box of Figure 7a, cars are shadowed, making the segmentation task more challenging. MFMamba, in this case, causes severe connection between the cars, while our MMFNet method successfully segments the ground between the cars, demonstrating its robustness in mitigating shadow-related challenges.

#### 4.4.2. Comparison Results on the Potsdam Dataset

As shown in Table 2, the experimental results on the Potsdam dataset are consistent with those on the Vaihingen dataset, where our method achieves the highest scores in terms of OA, mF1, and mIoU. Compared with the baseline MFMamba, our method improves OA, mF1, and mIoU by 0.43%, 0.39%, and 0.65%, respectively. Notably, it also demonstrates superior segmentation performance in key categories such as building, tree, low vegetation, and impervious surface when compared with other SOTA methods. Specifically, our method yields an improvement of 1.45% in IoU for low vegetation and 0.81% in IoU for buildings over the baseline.

Figure 8 presents visual examples of the segmentation results produced by all nine methods on the Potsdam dataset. Although the performance differences in terms of OA and mIoU are not particularly large, our method shows noticeably better performance in segmenting large buildings and visually confusing regions compared to other networks. As shown in the top-right rectangular windows of the first and third rows, MMFNet is able to accurately identify the entire building structure, whereas other methods produce fragmented results or mistakenly classify other objects as buildings. From the purple boxes in the second and third rows, it is evident that our method performs better in confusing areas than the others. In the second row, the road highlighted in the rectangular window is narrow and surrounded by dense vegetation, making it easy to be confused with the surrounding vegetation. MMFNet successfully distinguishes the road from the vegetation, while other methods tend to confuse the two. Low vegetation and trees are also commonly confused due to their inter-class similarity. In the bottom-left purple box of the third row, where trees are surrounded by low vegetation, MMFNet is able to accurately separate the two categories, whereas other methods tend to confuse them. Our method also demonstrates better performance in texture-missing scenarios. As shown in the purple box in Figure 7b, where the road texture is missing, our method successfully identifies the texture-missing area as a road, while MFMamba incorrectly classifies it as clutter.

#### 4.4.3. Computational Complexity Analysis

We adopt floating point operations (FLOPs) and the number of model parameters as evaluation metrics to assess the computational complexity of the proposed MMFNet. FLOPs serve as an indicator of the time complexity of deep learning-based models, while the parameter count quantifies model size. Table 3 presents the complexity analysis results for all comparison methods in this study.

As shown in Table 3, although UnetFormer has the lowest FLOPs and parameter count, its mIoU score is significantly lower than that of our model. Compared with Transformer-based multimodal methods such as CMFNet, MMFNet achieves a substantial reduction in FLOPs and requires fewer parameters while maintaining superior mIoU performance. This efficiency is primarily attributed to the use of Mamba as the auxiliary branch in the encoder, in contrast to Transformer, which is typically more resource-intensive.

When compared with MFMamba, another multimodal method based on RS3Mamba, MMFNet has a larger number of parameters, but the FLOPs remain the same, and the segmentation performance is significantly improved. Compared to single-modality segmentation methods, the computational complexity of our model is slightly higher due to the incorporation of multimodal data, yet it achieves notably better segmentation performance. Furthermore, relative to the baseline RS^3^Mamba, MMFNet yields substantial improvements in segmentation accuracy with only a modest increase in the number of model parameters.

### 4.5. Ablation Studies

To evaluate the effectiveness of incorporating DSM data, we conduct ablation experiments by setting the input to HRRSI only and HRRSI + DSM on Vaihingen and Potsdam datasets. As shown in Table 4, the inclusion of DSM data leads to overall performance improvements of MMFNet on both datasets. In particular, the most significant gains are observed in the segmentation of impervious surfaces and buildings, which can be attributed to the stable elevation characteristics of these classes.

However, the segmentation accuracy for low vegetation and trees shows slightly different trends across the two datasets. Specifically, the inclusion of DSM improves the performance of MMFNet on low vegetation in the Vaihingen dataset but reduces its accuracy on trees. Conversely, in the Potsdam dataset, adding DSM decreases the performance on low vegetation but enhances the segmentation of trees. This may be due to the strong spectral similarity between these two classes, along with their highly irregular structures and ambiguous boundary shapes. These results suggest that while DSM contributes to improved overall segmentation performance, distinguishing between trees and low vegetation remains a challenge due to their inherent inter-class similarity and spatial complexity.

To validate the effectiveness of the proposed WCA module and the integration of FreqFusion, we conduct ablation experiments by comparing the model performance with different components added. The evaluation results are summarized in Table 5, where a tick (√) indicates that the corresponding module is included. The first row of Table 5 presents the ablation result for FreqFusion, where bilinear interpolation is used to upsample feature maps instead of the proposed frequency-aware method. The second row shows the ablation result for WCA. In this setting, the structure of MFFB is retained, but the fusion operation is modified: the local feature F_Local_ and the global feature F_Global_ are fused via element-wise addition without the WCA module.

The results in the first and third rows demonstrate that FreqFusion enables effective complementary fusion between deep and shallow features. The notable improvements in global semantics (e.g., building) and local details (e.g., low vegetation) suggest that this module enhances the model’s capability to represent multi-scale features.

The results in the second and third rows confirm that the WCA module improves semantic understanding in complex scenes by adaptively fusing depth and RGB features. The performance gains for confusing categories such as low vegetation indicate that WCA enhances the quality of cross-modal feature interaction.

Overall, the third-row results show that both components contribute positively to model accuracy, and their combination yields the best performance.

The results in the first and third rows demonstrate that FreqFusion enables complementary fusion between deep and shallow features. The significant improvements in global semantics (building) and local details (low vegetation) indicate that this module enhances the model’s ability to represent multi-scale features. The results in the second and third rows verify that the WCA module improves semantic understanding in complex scenes by adaptively fusing DSM and HRRSI features. The observed improvements in confusing categories such as low vegetation suggest that WCA enhances the quality of cross-modal feature interaction. Overall, the third row confirms that both components contribute positively to model accuracy, and their combination yields the best performance.

## 5. Conclusions

In this study, we proposed MMFNet, a Mamba-based MSS network for remote sensing, designed to address the challenges of complex scene understanding, multimodal feature fusion, and computational efficiency. By integrating the strengths of CNN and Mamba architectures, MMFNet effectively leverages high-resolution spectral information (HRRSI) and DSM data to improve segmentation accuracy while maintaining low computational complexity. To better fuse multimodal features, we introduced the MFFB, which employs a window-based cross-attention mechanism to achieve adaptive fusion across modalities. This design effectively alleviates the insufficient interaction between global and local information in traditional approaches. Additionally, a frequency-aware upsampling module is incorporated into the decoder to reduce the loss of spatial detail during conventional upsampling and to facilitate semantic fusion of deep and shallow features, thereby enhancing edge segmentation accuracy. Compared to existing CNN- or Transformer-based methods, MMFNet offers a novel perspective for semantic segmentation of multimodal RSI. Experimental results on two public benchmarks ISPRS Vaihingen and Potsdam demonstrate that MMFNet outperforms eight state-of-the-art methods in segmentation performance, while maintaining relatively low computational cost.

Nonetheless, due to differences in imaging mechanisms, feature misalignment between DSM and RGB (IRRG) data can lead to misclassification in the segmentation results. In future work, we plan to explore cross-modal feature alignment techniques to better exploit complementary information across modalities, further improving the segmentation accuracy of HSRSI. To this end, we also plan to further extend MMFNet by integrating semantic segmentation with 3D scene reconstruction and object measurement tasks within a Multi-Task Learning (MTL) framework. MTL leverages shared knowledge among related tasks to enhance the generalization ability of each individual task. Its primary advantage lies in promoting cross-task feature sharing, which enables the model to learn more robust and transferable representations. By jointly optimizing multiple objectives, MTL can reduce the risk of overfitting to a single task while improving the overall efficiency and effectiveness of learning. This integration will leverage semantic segmentation for detailed object classification while providing structured spatial context for 3D scene reconstruction, resulting in more accurate geometric details. This unified approach will enable precise object identification and classification, ultimately enhancing the quality of 3D modelling and measurement. Furthermore, we aim to apply MMFNet to more recent multimodal benchmarks, including hyperspectral-DSM and SAR-DSM datasets, to validate its generalization capabilities beyond the ISPRS benchmarks.

MMFNet demonstrates strong potential for real-world remote sensing applications. Its semantic segmentation capability can support land cover classification in urban planning and environmental monitoring. The model’s robustness in identifying shadow-occluded roads and buildings, along with its ability to distinguish spectrally similar classes such as low vegetation and trees, makes it well-suited for complex scenes with challenging illumination and high inter-class similarity. Nevertheless, MMFNet exhibits limited performance in segmenting small-scale objects, and in steep terrain areas, elevation-induced feature distortion may lead to misclassification of roads as buildings. Recognizing these strengths and limitations is essential for guiding its deployment in operational scenarios—including disaster mapping, urban infrastructure monitoring, and ecological surveys—where appropriate preprocessing or model adaptation could further enhance its practical performance.

## Figures and Tables

**Figure 1 sensors-25-06225-f001:**
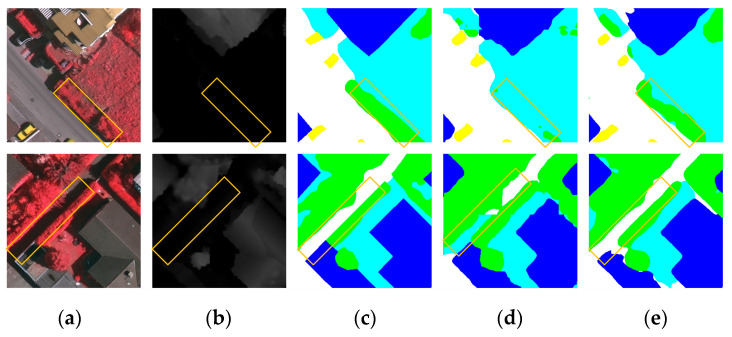
Predictions for objects with similar colour and texture. (**a**) RGB image; (**b**) Digital Surface Model (DSM); (**c**) ground truth (GT); (**d**) prediction using unimodal data; (**e**) prediction using multimodal data.

**Figure 2 sensors-25-06225-f002:**
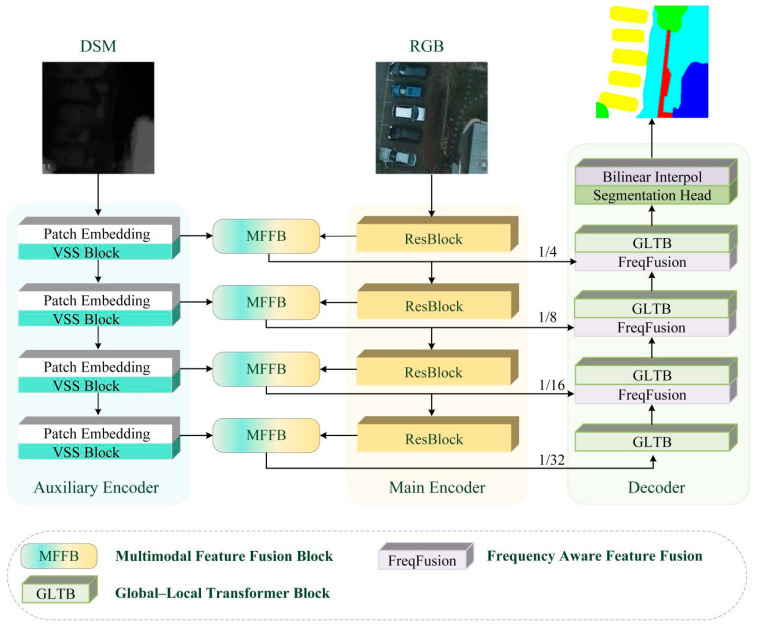
Overall architecture of MMFNet.

**Figure 3 sensors-25-06225-f003:**
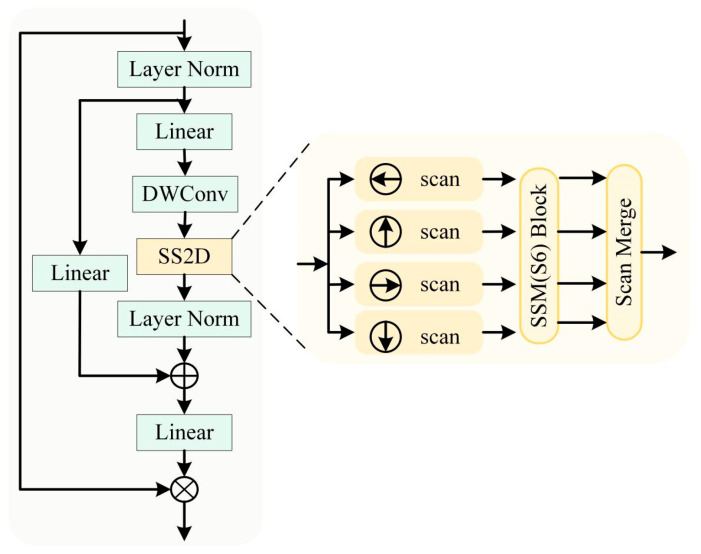
Overall structure of the Visual State Space (VSS) module.

**Figure 4 sensors-25-06225-f004:**
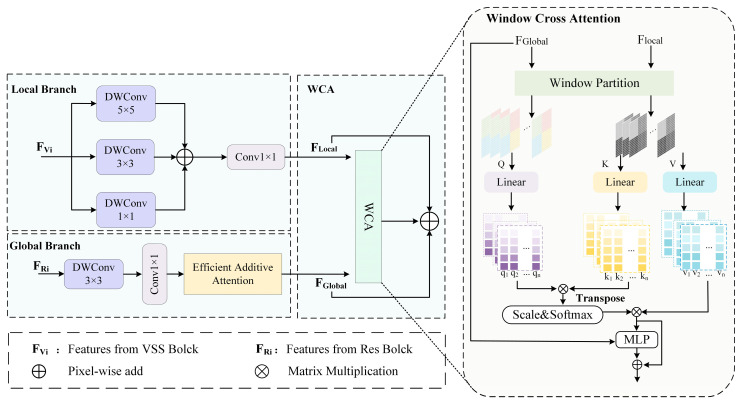
Overall architecture of the Multimodal Feature Fusion Block (MFFB).

**Figure 5 sensors-25-06225-f005:**
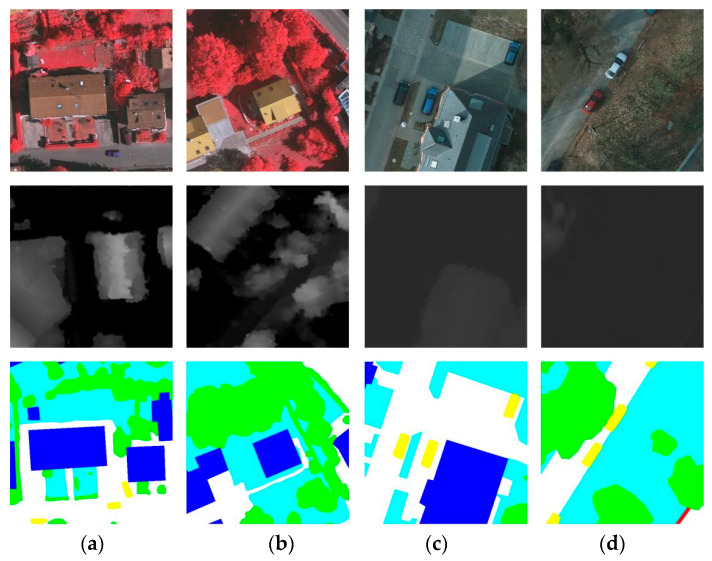
(**a**,**b**) are sample images from the Vaihingen dataset; (**c**,**d**) are from the Potsdam dataset. The first row shows the orthophotos with three spectral channels (NIRG for Vaihingen and RGB for Potsdam). The second and third rows present the corresponding DSM data and pixel-wise semantic labels, respectively.

**Figure 6 sensors-25-06225-f006:**
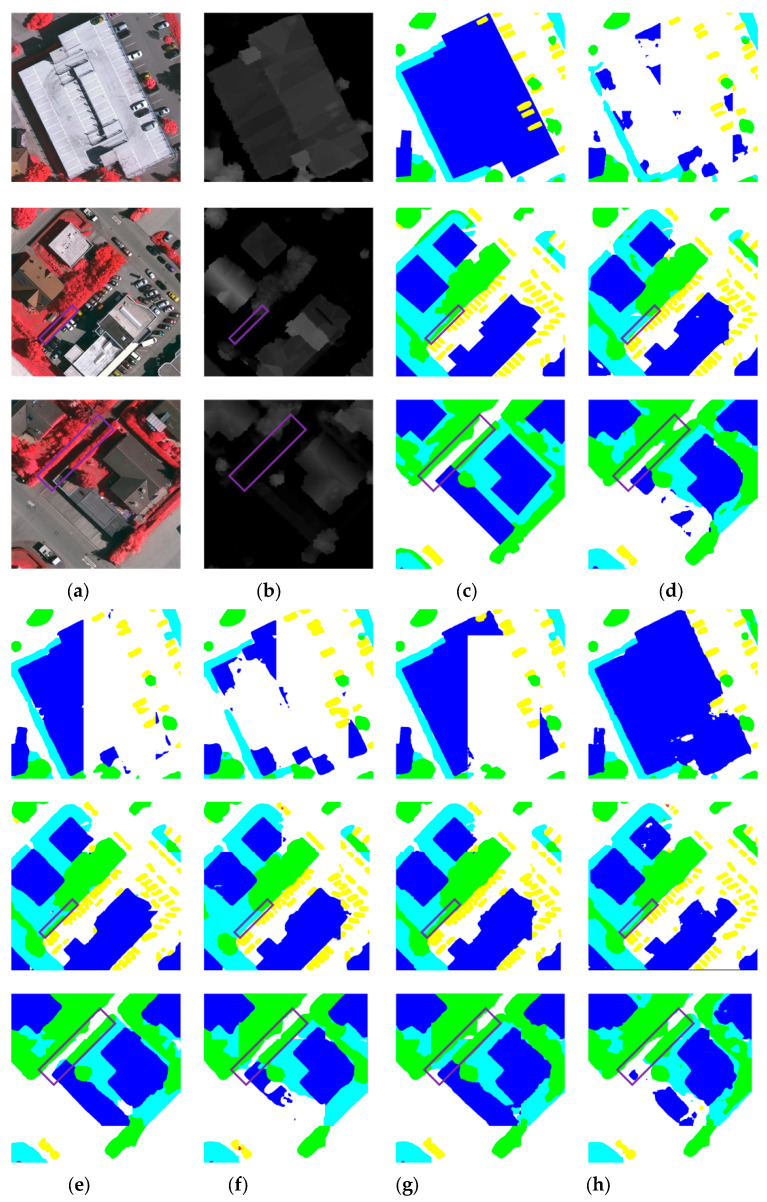
Visual comparison of segmentation results from different methods and the original data on the Vaihingen dataset. (**a**) NIRRG; (**b**) DSM; (**c**) GT; (**d**) PSPNet; (**e**) Swin; (**f**) UNetFormer; (**g**) DCSwin; (**h**) CMFNet; (**i**) VMamba; (**j**) RS3Mamba; (**k**) MFMamba; (**l**) Ours.

**Figure 7 sensors-25-06225-f007:**
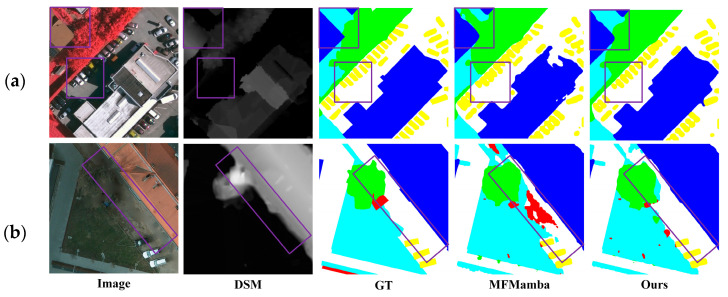
Comparison with baseline (MFMamba) in Difficult Segmentation Regions. (**a**) Example on Vaihingen Dataset; (**b**) Example on Potsdam Dataset.

**Figure 8 sensors-25-06225-f008:**
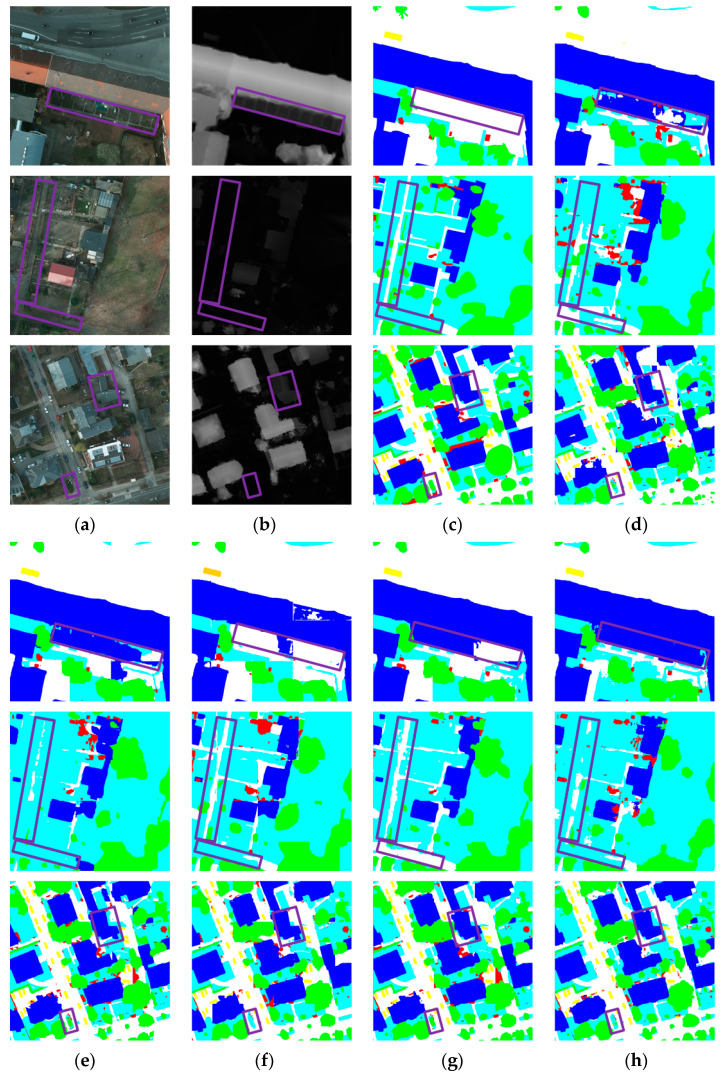
Visual comparison of segmentation results from different methods and the original data on the Potsdam dataset. (**a**) RGB; (**b**) DSM; (**c**) GT; (**d**) PSPNet; (**e**) Swin; (**f**) UNetFormer; (**g**) DCSwin; (**h**) CMFNet; (**i**) VMamba; (**j**) RS3Mamba; (**k**) MFMamba; (**l**) Ours.

**Table 1 sensors-25-06225-t001:** Comparison results with other methods on the Vaihingen dataset. Bold numbers indicate the optimal value, and underlined numbers represent the sub-optimal value. (Unit: %).

Model	Backbone	Imp.	Bui.	Low.	Tre.	Car	OA	mF1	mIoU
		IoU							
PSPNet	Resnet-18	79.27	89.5	60.41	77.25	71.45	87.58	86.31	75.76
Swin	Swin-T	81.49	89.93	63.08	75.05	64.97	86.74	84.87	74.90
Unetformer	Resnet-18	79.33	88.67	61.86	73.56	70.73	86.65	85.31	74.83
DCSwin	Swin-T	81.47	89.81	63.69	74.82	70.54	87.70	86.11	76.07
CMFNet	VGG-16	86.59	94.25	66.75	82.75	77.03	91.39	89.50	81.47
Vmamba	Vmamba-T	82.58	89.06	67.35	77.83	57.07	82.23	84.56	74.78
RS3Mamba	R18-Mamba-T	85.21	92.34	66.51	82.94	81.21	90.87	89.28	81.64
MFMamba	R18-Mamba-T	86.04	94.02	66.14	83.64	77.43	91.37	89.48	81.46
Ours	R18-Mamba-T	**87.55**	**94.37**	**68.92**	**84.21**	**82.42**	**92.06**	**90.77**	**83.50**

**Table 2 sensors-25-06225-t002:** Comparison results with other methods on the Potsdam dataset (Unit: %). Bold numbers indicate the optimal value, and underlined numbers represent the sub-optimal value.

Model	Backbone	IoU					OA	mF1	mIoU
		Imp.	Bui.	Low.	Tre.	Car			
PSPNet	Resnet-18	78.98	88.93	68.23	68.42	77.77	86.12	82.51	76.47
Swin	Swin-T	79.28	90.5	69.98	70.41	79.64	87.05	83.69	77.96
Unetformer	Resnet-18	84.51	92.08	72.70	71.42	83.45	89.19	89.20	80.83
DCSwin	Swin-T	82.96	92.50	71.31	71.24	82.29	88.31	88.71	80.06
CMFNet	VGG-16	85.55	93.65	72.23	74.65	91.25	89.97	91.01	83.37
Vmamba	Vmamba-T	84.82	91.24	75.16	75.38	88.04	81.52	90.40	82.93
RS3Mamba	R18-Mamba-T	86.95	94.46	75.50	76.28	92.98	90.73	87.39	85.24
MFMamba	R18-Mamba-T	87.34	94.90	75.09	76.81	92.88	90.89	91.92	85.41
Ours	R18-Mamba-T	**87.41**	**95.71**	**76.54**	**77.50**	**93.13**	**91.32**	**92.31**	**86.06**

**Table 3 sensors-25-06225-t003:** Comparison of Computational Complexity and mIoU Performance on the Vaihingen Dataset.

Method	FLOPs (G)	Parameter (M)	mIoU (%)
PSPNet	64.15	65.60	75.76
Swin	60.28	59.02	74.90
Unetformer	5.99	11.72	74.83
DCSwin	40.08	66.95	76.07
CMFNet	159.55	104.07	81.47
Vmamba	12.41	29.94	74.78
RS3Mamba	19.78	43.32	81.64
MFMamba	19.12	62.43	81.46
Ours	19.15	69.85	83.50

**Table 4 sensors-25-06225-t004:** Ablation Study on the Effect of DSM Data on the Vaihingen and Potsdam Datasets.

Dataset	Bands	Class OA (%)					mF1 (%)	mIoU (%)
		Imp.	Bui.	Low.	Tre.	Car		
Vaihingen	NIRRG	92.04	96.06	80.64	92.04	88.36	90.04	82.29
	NIRRG + DSM	93.57	97.21	81.16	91.50	88.39	90.77	83.50
Potsdam	RGB	92.45	97.50	88.94	86.63	96.11	91.63	84.91
	RGB + DSM	92.70	98.01	88.78	87.65	96.38	92.31	86.06

**Table 5 sensors-25-06225-t005:** Ablation Study of WCA and FreqFusion on the Potsdam Dataset (Unit: %).

WCA	FreqFusion	Imp.	Bui.	Low.	Tre.	Car	OA	mF1	mIoU
√	×	87.53	95.49	75.71	77.48	93.30	91.13	92.21	85.90
×	√	87.32	95.31	75.44	77.50	93.09	90.97	92.12	85.73
√	√	87.41	95.71	76.54	77.50	93.13	91.32	92.31	86.06

## Data Availability

The Vaihingen and the Potsdam datasets can be obtained from https://www.isprs.org/resources/datasets/benchmarks/UrbanSemLab/Default.aspx (accessed on 1 October 2024).

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
