# Peer review of "MMFNet: A Mamba-Based Multimodal Fusion Network for Remote Sensing Image Semantic Segmentation"

_sensors, 2025, doi:10.3390/s25196225_

Round 1

Reviewer 1 Report

Comments and Suggestions for Authors

This manuscript presents MMFNet, a Mamba-based multimodal fusion network for semantic segmentation of high-resolution remote sensing images, integrating RGB (or NIRRG) and DSM data. The work addresses two important challenges in multimodal semantic segmentation (MSS): (1) balancing segmentation accuracy and computational efficiency, and (2) achieving effective feature fusion between heterogeneous modalities. The authors propose a dual-encoder architecture: ResNet-18 for local detail extraction and VMamba for global context modeling, together with a Multimodal Feature Fusion Block (MFFB) and a frequency-aware guided upsampling module (FreqFusion). Experiments on the ISPRS Vaihingen and Potsdam datasets show consistent improvements over eight state-of-the-art methods in both mIoU and mF1, with relatively low computational complexity.

Strengths

  1. Clear Motivation and Novelty – The paper clearly identifies the limitations of CNN- and Transformer-based multimodal segmentation and motivates the use of Mamba to capture long-range dependencies with linear complexity. The integration of MFFB and FreqFusion addresses well-known fusion and boundary delineation issues.
  2. Methodological Soundness – The proposed MFFB’s dual-branch design (global branch with Efficient Additive Attention, local branch with multi-kernel depthwise convolutions, window-based cross-attention) is technically coherent and well justified. FreqFusion is appropriately adapted to enhance fine-grained spatial details.
  3. Comprehensive Experiments – Results on two benchmark datasets, comparison with multiple baselines (including both CNN, Transformer, and Mamba variants), computational complexity analysis, and ablation studies provide solid empirical support for the claims.
  4. Performance Gains – The method achieves noticeable improvements in challenging categories such as low vegetation vs. trees and in shadowed regions, which are common pitfalls for unimodal and simple fusion methods.

Weaknesses / Points for Improvement

  1. Some parts of the paper contain repetitive and redundant expressions, and it is recommended to streamline and refine them.

(1) In the Abstract, when introducing the MFFB module, the focus should be more on its role and effectiveness rather than specific technical details, which can be described only briefly.

(2) In the Introduction, the detailed descriptions of HRRSI challenges in Page 1, Lines 39–40 and Lines 41–42, as well as Page 2, Lines 45–48, are repetitive and can be streamlined into a single paragraph, focusing only on the motivation.

(3) In the Proposed Method section, there is redundancy and overlap in the logical structure between Sections 3.1 (Framework) and 3.2 (Encoder). It is recommended to streamline the content of Sections 3.1 and 3.2, with Section 3.1 focusing more on the overall network architecture and Section 3.2 providing a detailed description of the encoder’s structural details.

  1. In the Experiments and Results section, table 1 and table 2 captions state “Bold numbers indicate the optimal value, and underlined numbers represent the sub-optimal value,” but these markings are not shown in the tables.
  2. The line formatting in Table 4 is not standardized and should be revised.
  3. Insufficient reproducibility and detail, with a lack of complete descriptions regarding training specifics (optimizer parameters, learning rate schedule, and data augmentation strategy). It is recommended to provide a GitHub link.
  4. The paper focuses more on algorithm performance, while lacking discussion of its application in real remote sensing sensor systems or practical tasks (e.g., land cover classification, disaster monitoring). It is recommended to add an analysis of potential application scenarios in the conclusion or discussion section to enhance its practical engineering value.

Reviewer 2 Report

Comments and Suggestions for Authors

Reviewer 3 Report

Comments and Suggestions for Authors

Remarks for authors.

  1. Please estimate statistical significance of the results obtained, where it is possible.
  2. The authors can also analyze the misclassification, which appear in experiments with the databases. For clarity, it is worth showing some fragments of images in close-up, enlarged form.
  3. The first paragraph of section 2.2 (p.4), that begins with “The Materials and Methods should be described…”  must be removed from the manuscript, because it is obviously copied from the authors instructions.
  4. Ensure that the figures match the text. Figure 7 is titled as “Visual comparison of segmentation results…”, but columns (a), (b) contain original data, not  segmentation results.
  5. Check grammar/consistency: “long range” vs. “long‑range”, ”multimodal” vs. ”multimodel” etc.
  6. For future research, I recommend to integrate semantic segmentation with 3-D scene reconstruction and measurement of the objects observed.

Round 2

Reviewer 2 Report

Comments and Suggestions for Authors
